# Prediction of recurrent stroke among ischemic stroke patients with atrial fibrillation: Development and validation of a risk score model

**Beom Joon Kim**[1], **Keon-Joo Lee**[1], **Eun Lyeong Park**[2], **Kanta Tanaka**[3], **Masatoshi Koga**[4], **Sohei Yoshimura**[4], **Ryo Itabashi**[5], **Jae-Kwan Cha**[6], **Byung-Chul Lee**[7], **Hisanao Akiyama**[8], **Yoshinari Nagakane**[9], **Juneyoung Lee**[2], **Kazunori Toyoda**[4], **for the SAMURAI Study Investigators**[¶], **Hee-Joon Bae**[1]*, **for the CRCS-K Investigators**[¶]

1 Department of Neurology and Cerebrovascular Center, Seoul National University Bundang Hospital, Seoul National University College of Medicine, Seongnam-si, Republic of Korea, 2 Department of Biostatistics, College of Medicine, Korea University, Seoul, Republic of Korea, 3 Division of Stroke Care Unit, National Cerebral and Cardiovascular Center, Suita, Japan, 4 Department of Cerebrovascular Medicine, National Cerebral and Cardiovascular Center, Suita, Japan, 5 Department of Stroke Neurology, Kohnan Hospital, Sendai, Japan, 6 Department of Neurology, Dong-A University Hospital, Busan, Republic of Korea, 7 Department of Neurology, Hallym University Sacred Heart Hospital, Anyang, Korea, 8 Division of Neurology, Department of Internal Medicine, St Marianna University of Medicine, Kawasaki, Japan, 9 Department of Neurology, Kyoto Second Red Cross Hospital, Kyoto, Japan

¶ Membership of the SAMURAI Study and CRCS-K are listed in the Acknowledgments.
* braindoc@snu.ac.kr

**Data Availability Statement:** Secondary use of the registry data and additional review of medical

## Abstract

### Background

There is currently no validated risk prediction model for recurrent events among patients with acute ischemic stroke (AIS) and atrial fibrillation (AF). Considering that the application of conventional risk scores has contextual limitations, new strategies are needed to develop such a model. Here, we set out to develop and validate a comprehensive risk prediction model for stroke recurrence in AIS patients with AF.

### Methods

AIS patients with AF were collected from multicenter registries in South Korea and Japan. A developmental dataset was constructed with 5648 registered cases from both countries for the period 2011–2014. An external validation dataset was also created, consisting of Korean AIS subjects with AF registered between 2015 and 2018. Event outcomes were collected during 1 year after the index stroke. A multivariable prediction model was developed using the Fine–Gray subdistribution hazard model with non-stroke mortality as a competing risk. The model incorporated 21 clinical variables and was further validated, calibrated, and revised using the external validation dataset.

### Results

The developmental dataset consisted of 4483 Korean and 1165 Japanese patients (mean age, 74.3 ± 10.2 years; male 53%); 338 patients (6%) had recurrent stroke and 903 (16%) died. The

records for the current study were approved by the IRB of Seoul National University Bundang Hospital [B-1705/396-306]. The source data could not be made publicly available due to legal constraints, specifically the Personal Information Protection Act (2014). No explicit informed consent for public archiving of the pseudonymized source data has been obtained, in which case local regulations preclude public archiving of the data. The pseudonymized data that support the findings of this study are available from the corresponding author, Dr. Hee-Joon Bae, or the IRB of Seoul National University Bundang Hospital (82, Gumi-ro 173 Beon-gil, Bundang-gu, Seongnam-si, Gyeonggi-do 13605, South Korea; https://msri. snubh.org) upon reasonable request, subsequent approval from the local IRB, and completion of a legal data sharing agreement.

**Funding:** This study was supported by Bristol–Myers Squibb Korea and the Korea Centers for Disease Control and Prevention (no. 2020ER620200#) granted to HJB and a Grant-in-Aid (H23-Junkanki-Ippan-010) from the Ministry of Health, Labour and Welfare, Japan granted to KT. The funding sources did not participate in any part of the study, from conception to article preparation.

**Competing interests:** This study was supported by Bristol–Myers Squibb Korea and the Korea Centers for Disease Control and Prevention (no. 2020ER620200#) and a Grant-in-Aid (H23-Junkanki-Ippan-010) from the Ministry of Health, Labour and Welfare, Japan. This does not alter our adherence to PLOS ONE policies on sharing data and materials. The funding sources did not participate in any part of the study, from conception to article preparation.

clinical profiles of the external validation set (n = 3668) were comparable to those of the developmental dataset. The c-statistics of the final model was 0.68 (95% confidence interval, 0.66 –0.71). The developed prediction model did not show better discriminative ability for predicting stroke recurrence than the conventional risk prediction tools (CHADS$_2$, CHA$_2$DS$_2$-VASc, and ATRIA).

## Conclusions

Neither conventional risk stratification tools nor our newly developed comprehensive prediction model using available clinical factors seemed to be suitable for identifying patients at high risk of recurrent ischemic stroke among AIS patients with AF in this modern direct oral anticoagulant era. Detailed individual information, including imaging, may be warranted to build a more robust and precise risk prediction model for stroke survivors with AF.

## Introduction

Atrial fibrillation (AF) is a well-known risk factor for systemic embolic events, including ischemic stroke [1]. Nonvalvular AF independently increases the risk of stroke by almost five-fold across all age-groups [2]. The excess event rate of stroke due to AF was estimated to be 10.4/ 1000 person-years in middle-aged and 18.3/1000 person-years in older individuals in a Japanese cohort study [3]. Anticoagulation with warfarin or direct oral anticoagulants (DOACs) has been proven to reduce the risk of recurrent stroke and systemic embolization [4–7]. However, considering the potential risk of bleeding complications, it is necessary to weigh the benefits and risks from anticoagulation before initiating treatment.

Various risk stratification tools to predict stroke in non-valvular AF patients have been developed and are widely used in clinical practice; these include the CHADS$_2$ score, CHA$_2$DS$_2$-VASc score, and ATRIA score [8–10]. However, these scores have limited applicability in treatment decisions for patients with acute ischemic stroke (AIS) and AF. The score schemas were developed from community-based cohorts; thus, stroke survivors were rare in the developmental datasets of these tools. Furthermore, the mainstay treatment for secondary prevention at the time when these scores were developed was vitamin K antagonists; hence, the validity of their usage in the modern DOAC era is questionable. Recent advances in electronic health record systems and stroke imaging make it possible to obtain the ample information that is required to choose an antithrombotic strategy in patients with AIS. AF patients may suffer ischemic stroke despite antithrombotic mediation, and such cases have an elevated risk of recurrent stroke [11, 12]. Moreover, both the risks of ischemic and hemorrhagic strokes were numerically higher in the Asian population [13–16]. which were not adequately represented in the developmental datasets of the existing risk stratification tools.

Considering that the application of conventional risk scores is limited by the context of their clinical milieu and developmental dataset, a whole new set of developmental strategies may be required in developing a new stroke recurrence prediction tool for AIS patients with AF. In this study, we developed and validated a comprehensive risk prediction model for recurrent strokes using prospective stroke registries from South Korea and Japan.

## Methods

### Study subjects and clinical data collection

This study was a retrospective analysis of prospectively collected databases from multicenter registries in South Korea and Japan. AIS patients with documented non-valvular AF who were

hospitalized between 2011 and 2014 were identified from the Clinical Research Collaboration for Stroke in Korea (CRCS-K; n = 4844) and the Stroke Acute Management with Urgent Risk-factor Assessment and Improvement (SAMURAI)-NVAF study (n = 1192) [17, 18]. Among the 6036 collected patients, 388 patients were excluded due to in-hospital death (n = 385) or a lack of outcome information (n = 3). A total of 5648 patients were thus included in the developmental dataset. The external validation dataset was comprised of 3668 AIS patients with non-valvular AF who were hospitalized between 2015 and 2018 and were registered in the CRCS-K. The developmental and external validation datasets were mutually exclusive (Fig 1).

The data dictionaries and elements were harmonized to generate a comparable and interchangeable common dataset using the CRCS-K and SAMURAI-NVAF databases. The common dataset included demographic data, baseline clinical profiles, stroke information, laboratory information, in-hospital treatments, discharge medications, and outcome data. Functional outcomes were modified Rankin Scale scores at 3 months and at 1 year after the index stroke. Recurrent stroke, myocardial infarction, and death for up to 1 year were collected as event outcomes. All the information recorded in the source databases was retrieved to construct a common dataset. All the study participants or their next of kin had given written consents to participate in the CRCS-K or SAMURAI-NVAF studies. The local institutional review

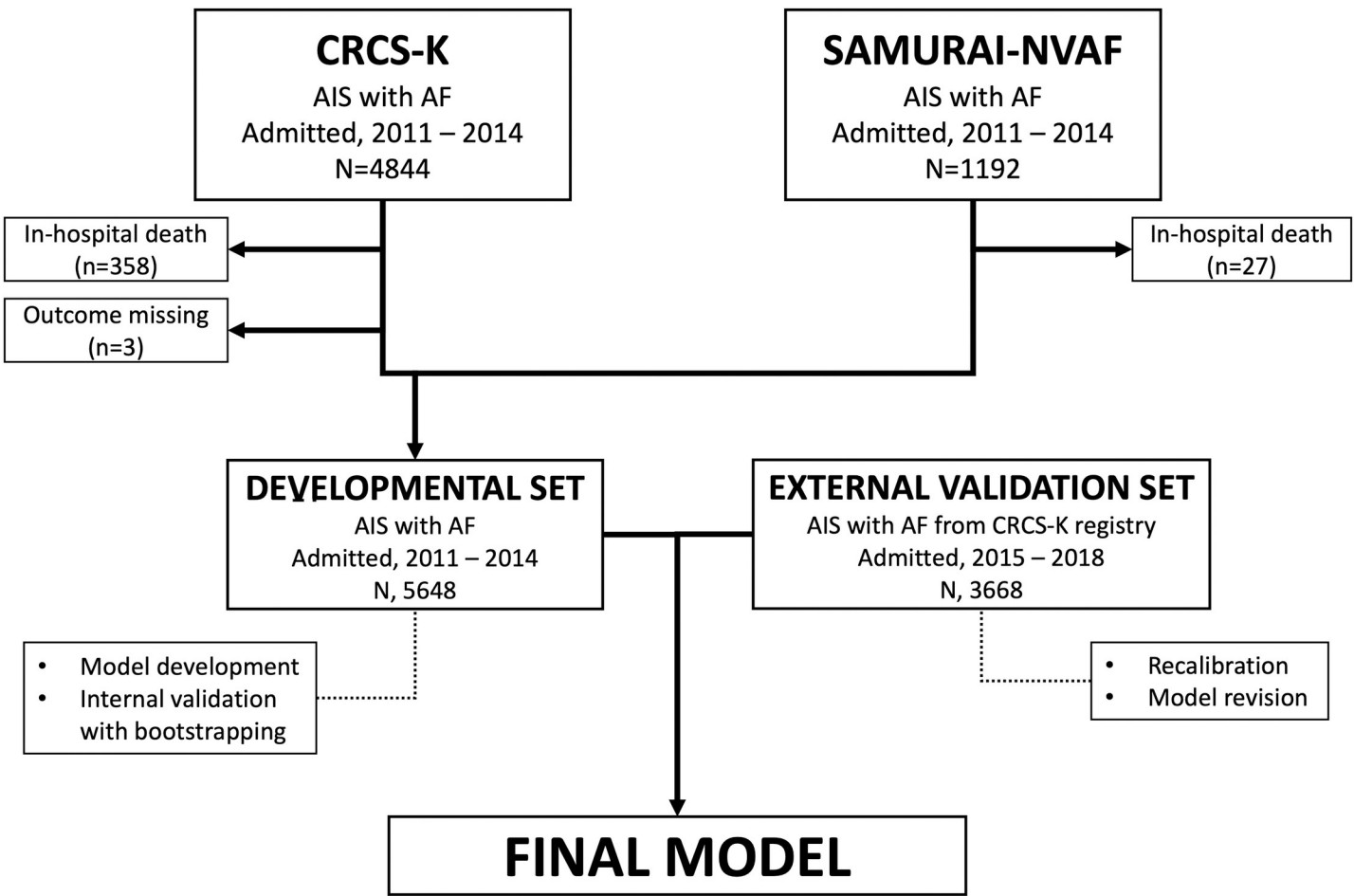

**Fig 1. Overview of the study design.**

boards (IRBs) of all participating centers approved the original CRCS-K and SAMURAI-N-VAF study. Secondary use of the registry data and additional review of medical records for the current study were approved by the IRB of Seoul National University Bundang Hospital [B-1705/396-306]. The source data could not be made publicly available due to legal constraints, specifically the Personal Information Protection Act (2014). No explicit informed consent for public archiving of the pseudonymized source data has been obtained, in which case local regulations preclude public archiving of the data. The pseudonymized data that support the findings of this study are available from the corresponding author, Dr. Hee-Joon Bae, or the IRB of Seoul National University Bundang Hospital (82, Gumi-ro 173 Beon-gil, Bundang-gu, Seong-nam-si, Gyeonggi-do 13605, South Korea; https://msri.snubh.org) upon reasonable request, subsequent approval from the local IRB, and completion of a legal data sharing agreement.

## Development and validation of the prediction model for recurrent stroke

A risk prediction model for recurrent stroke was developed and validated according to published guidelines [19, 20]. Potential predictors for recurrent stroke were retrieved from the developmental dataset. Candidate variables were selected based on published evidence, clinical experience, and the availability of data elements. Variables related to antithrombotic medications at discharge (anticoagulants and antiplatelet agents) were included in the models. Selected variables were checked for missing data, multicollinearity, influential observations, and goodness-of-fit in the models. For explanatory variables whose relationships with the outcome variable (logarithm of time to recurrent stroke) were nonlinear, appropriate transformation was made based on Akaike's information criteria to maximize the predictability of the model. Finally, a multivariable model incorporating significant interaction terms between predictor variables was developed using the Fine–Gray subdistribution hazard model. Non-stroke mortality was considered a competing event (n = 903).

For internal validation, regression parameter estimates were re-estimated with the bootstrapping method, in which the whole dataset was sampled using 999 repetitions with replacement [21]. Measures used to examine the model's predictive performance were Harrel's c-statistic for discrimination ability, Nagelkerke's $R^2$ for variation explained, and a discrimination slope for agreement between predicted and observed probabilities.

External validation was performed to calibrate and revise the regression coefficients of the developed model, using an independently collected dataset of the 3668 AIS patients with AF. The overall slope was calibrated by refitting a null model using the linear predictors of the developed model as an offset variable. Next, for each of the variables with p-values less than 0.5, the regression parameter was revised according to the method described previously.[20] To examine the performance of the final prediction model, the model's predicted risks were categorized into deciles, and their percent prediction for recurrent stroke was compared with the event proportions according to conventional $CHADS_2$, $CHA_2DS_2$-VASc, and ATRIA scores in both the developmental and external validation datasets. Due to the lack of information on proteinuria in the developmental and external validation datasets, it was randomly imputed with a Bernoulli (p = 0.5) distribution.

Baseline characteristics were summarized as frequencies (percentages), mean ± standard deviation (SD), or median (interquartile range, IQR), as appropriate. Differences between categories were evaluated using the chi-squared test or Student's *t*-test. A Fine–Gray subdistribution hazard model was used to estimate the cumulative incidence of recurrent stroke. The significance level was set at a two-tailed p-value of < 0.05. All statistical analyses were performed using SAS version 9.4 (SAS Inc., Cary, NC, USA).

## Results

The 5648 AIS patients with documented non-valvular AF that were included in the developmental dataset were recruited from South Korea (n = 4483; 79%) and Japan (n = 1165; 21%). Mean age was 74 years, and 53.1% were male (Table 1). Vascular risk factors including hypertension, diabetes, dyslipidemia, and smoking were prevalent in this population. Before the index stroke, 35.5% of patients used antiplatelet medications and 20% used anticoagulants. Intravenous thrombolysis was administered in 16.2% of patients and endovascular recanalization treatment was administered in more than 10%. Recurrent stroke affected 6.0% of patients, but 16.0% died during the first year after the index stroke.

The clinical profiles of the included subjects differed by country. Japanese patients were more likely to be older, on anticoagulants prior to the index stroke, and less likely to be smokers. In the developmental dataset, a prescription of DOACs at the time of discharge was more frequent in the SAMURAI-NVAF dataset (40%) than in the CRCS-K database (3%). The median values of $CHADS_2$, $CHA_2DS_2$-VASc, and ATRIA scores were 4 (IQR, 3–4), 5 (4–6), and 9 (9–10), and their distributions were numerically comparable between the two countries (S1 and S2 Figs).

The external validation dataset was constructed using AIS patients with non-valvular AF who were hospitalized and registered in the CRCS-K between 2015 and 2018. Their clinical profiles were generally comparable to those of Korean patients in the developmental dataset. However, the frequency of DOAC prescription at discharge had increased to 49% in the external validation dataset (S1 Table).

We constructed a clinical prediction model for the risk of recurrent stroke among stroke survivors with non-valvular AF, treating all-cause mortality as a competing risk. The prediction model, incorporating the appropriately transformed variables and significant interaction terms, underwent internal validation through 999 bootstrap samples. We performed further calibration and revision of the model through the external validation dataset (Fig 2; S1 File). The final model is presented in Table 2.

The final model showed modest performance in predicting recurrent stroke, as assessed by the c-index (0.68 [95% CI, 0.66–0.71]). Table 3 and Fig 3 show the event rates of recurrent stroke for each of the currently available risk scores as well as the deciles of our prediction model, based on the developmental dataset of 5648 AIS patients with non-valvular AF. Neither the conventional risk scores nor our newly developed model showed a consistent dose-dependent relationship. The observed incidence rates of recurrent stroke according to the $CHADS_2$ and $CHA_2DS_2$-VASc scores dropped at the penultimate strata (5-point for $CHADS_2$ score and 7-point for $CHA_2DS_2$-VASc score). The incidence rate according to the ATRIA scores decreased in the higher score range. Our newly developed prediction model showed limited differentiation in the lower score range.

## Discussion

We built a clinical prediction model for recurrent stroke based on the 5648 AIS patients with non-valvular AF recruited from South Korea and Japan, using detailed clinical information that was easily collected during clinical practice. The model was further calibrated and revised using an external validation dataset. The comprehensive final model showed only modest utility in individual risk stratification, with similar performance as the conventional risk scores, such as $CHADS_2$, $CHA_2DS_2$-VASc, and ATRIA scores.

The model development and validation process adhered to the academic standards and published guidelines [19, 20]. Patient data were collected from two countries with different epidemiological characteristics and healthcare systems, to ensure the generalizability of the

**Table 1. Clinical profile of the developmental dataset, stratified by the recruiting countries.**

| | Whole population (N = 5648) | Korea (N = 4483) | Japan (N = 1165) | P-value |
|---|---|---|---|---|
| Age (years) | 74.3 ± 10.2 | 73.4 ± 10.1 | 77.5 ± 9.9 | <0.01 |
| Male | 2998 (53.1%) | 2340 (52.2%) | 658 (56.5%) | 0.01 |
| Onset to arrival (day) | | | | <0.01 |
| ≤ 24 hours | 3519 (62.3%) | 2722 (60.7%) | 797 (68.4%) | |
| 1–2 day | 1702 (30.1%) | 1412 (31.5%) | 290 (24.9%) | |
| ≥ 3 days | 427 (7.6%) | 349 (7.8%) | 78 (6.7%) | |
| Body mass index | 23.0 ± 3.6 | 23.2 ± 3.5 | 22.3 ± 3.8 | <0.01 |
| TIA as an index stroke | 162 (2.9%) | 111 (2.5%) | 51 (4.4%) | <0.01 |
| Pre-stroke mRS score ≥ 1 | 1253 (22.2%) | 1014 (22.6%) | 239 (20.5%) | 0.12 |
| NIHSS score | 8 [3–15] | 8 [3–15] | 8 [2–18] | 0.02 |
| Hypertension | 4132 (73.2%) | 3282 (73.2%) | 850 (73.0%) | 0.87 |
| Diabetes | 1509 (26.7%) | 1274 (28.4%) | 235 (20.2%) | <0.01 |
| Dyslipidemia | 1656 (29.3%) | 1272 (28.4%) | 384 (33.0%) | <0.01 |
| Smoking | 1491 (26.4%) | 1305 (29.1%) | 186 (16.0%) | <0.01 |
| Newly detected AF | 2425 (42.9%) | 1974 (44.0%) | 451 (38.7%) | <0.01 |
| Pre-stroke antiplatelets | 2005 (35.5%) | 1734 (38.7%) | 271 (23.3%) | <0.01 |
| Pre-stroke anticoagulation | 1129 (20.0%) | 754 (16.8%) | 375 (32.2%) | <0.01 |
| Arterial occlusions | | | | |
| ICA or MCA | 2311 (40.9%) | 1752 (39.1%) | 559 (48.0%) | <0.01 |
| Vertebro-basilar arteries | 263 (4.7%) | 204 (4.6%) | 59 (5.1%) | 0.46 |
| Others | 396 (7.0%) | 301 (6.7%) | 95 (8.2%) | 0.09 |
| Discharge medications | | | | |
| Aspirin | 1493 (26.4%) | 1412 (31.5%) | 81 (7.0%) | <0.01 |
| Clopidogrel | 535 (9.5%) | 487 (10.9%) | 48 (4.1%) | <0.01 |
| Cilostazol | 119 (2.1%) | 108 (2.4%) | 11 (0.9%) | <0.01 |
| Warfarin | 3568 (63.2%) | 2918 (65.1%) | 650 (55.8%) | <0.01 |
| Apixaban | 31 (0.5%) | 6 (0.1%) | 25 (2.1%) | <0.01 |
| Dabigatran | 265 (4.7%) | 62 (1.4%) | 203 (17.4%) | <0.01 |
| Rivaroxaban | 285 (5.0%) | 47 (1.0%) | 238 (20.4%) | <0.01 |
| Laboratory information | | | | |
| White blood cell count | 8077 ± 3029 | 8324 ± 3090 | 7128 ± 2571 | <0.01 |
| Hemoglobin | 13.4 ± 2.0 | 13.4 ± 2.0 | 13.4 ± 2.0 | 0.30 |
| Total cholesterol | 168 ± 39 | 165 ± 38 | 181 ± 37 | <0.01 |
| Creatinine | 1.03 ± 0.83 | 1.04 ± 0.80 | 1.03 ± 0.95 | 0.70 |
| Initial glucose at arrival | 137 ± 51 | 137 ± 51 | 136 ± 51 | 0.70 |
| Systolic blood pressure | 145 ± 26 | 142 ± 25 | 154 ± 27 | <0.01 |
| Diastolic blood pressure | 85 ± 16 | 84 ± 16 | 88 ± 19 | <0.01 |
| CHADS$_2$ score | 4 [3–4] | 4 [3–4] | 4 [3–4] | <0.01 |
| CHA$_2$DS$_2$-VASc score | 5 [4–6] | 5 [4–6] | 5 [4–6] | <0.01 |
| ATRIA score | 9 [9–10] | 9 [9–10] | 10 [9–11] | <0.01 |
| mRS score at discharge | | | | <0.01 |
| 0 | 790 (14.0%) | 593 (13.2%) | 197 (16.9%) | |
| 1 | 903 (16.0%) | 689 (15.4%) | 214 (18.4%) | |
| 2 | 849 (15.0%) | 700 (15.6%) | 149 (12.8%) | |
| 3 | 775 (13.7%) | 642 (14.3%) | 133 (11.4%) | |
| 4 | 1057 (18.7%) | 834 (18.6%) | 223 (19.1%) | |

(*Continued*)

**Table 1.** (Continued)

| | Whole population | Korea | Japan | P-value |
|---|---|---|---|---|
| | (N = 5648) | (N = 4483) | (N = 1165) | |
| 5 | 1274 (22.6%) | 1025 (22.9%) | 249 (21.4%) | |
| Recurrent stroke | 338 (6.0%) | 252 (5.6%) | 86 (7.4%) | <0.01 |
| F/U duration for stroke | 365 [247–365] | 365 [217–365] | 365 [344–365] | |
| Death up to 1 year | 903 (16.0%) | 774 (17.3%) | 129 (11.1%) | <0.01 |
| F/U duration (year) | 365 [331–365] | 365 [303–365] | 365 [365–365] | |

mRS, modified Rankin Scale; NIHSS, National Institute of Health Stroke Score; AF, atrial fibrillation; ICA, internal carotid artery; MCA, middle cerebral artery; F/U, follow-up.

final model. Competing risks from all-cause mortality were also incorporated into the final model, as AIS patients with non-valvular AF tend to have higher mortality during the first year after stroke [22]. We developed a comprehensive prediction model using 21 variables with appropriate variable transformation for linearity, if necessary, and four interaction terms. With advances in electronic health record keeping systems, automatic retrieval of the required data elements and calculation of a complex formula have become feasible in clinical practice. The developed model was validated and updated using a mutually exclusive external validation dataset. The final model's performance was compared to that of conventional risk prediction schemas.

The discrimination ability of our model appeared to be modest, with c-statistics of 0.68 [95% CI, 0.66–0.71]). This number was comparable to that of conventional risk scores. Based on 60594 UK patients with AF and without warfarin use, the c-statistics for conventional scores were 0.70 [0.69–0.71] for ATRIA, 0.68 [0.67–0.69] for CHADS$_2$, and 0.68 [0.67–0.69] for CHA$_2$DS$_2$-VASc scores [23]. In a Taiwanese National Healthcare Claims database study, the c-statistics were 0.70 [0.69–0.71] for CHA$_2$DS$_2$-VASc and 0.63 [0.62–0.64] for ATRIA scores [24].

Currently, there is no validated risk prediction tool for recurrent stroke among patients with non-valvular AF who survive the acute phase of ischemic stroke. Instead, the conventional risk scores are utilized even in patients who have already scored at least two points on the CHADS$_2$ and CHA$_2$DS$_2$-VASc risk schemas, and for whom, therefore, anticoagulation is automatically indicated. Considering the low risk of bleeding while on DOACs, it may be feasible to combine DOACs with antiplatelet therapy for patients with non-valvular AF and concomitant advanced atherosclerosis [25]. There is an urgent need to develop a new risk stratification tool for AIS patients with AF. However, the discrimination ability of both the newly developed model and conventional risk scores was unsatisfactory over the entire risk score strata. Overall, the risk prediction tools, including our newly developed model, showed modest performance in predicting recurrent stroke (Fig 3). There are irregularities that limit the applicability of these tools in clinical practice.

This unsatisfactory performance of the conventional tools and our newly developed model may be due to the following factors: First, ischemic stroke is a heterogeneous entity [26]. AF contributes strongly to the occurrence of ischemic stroke, but atherothrombosis or lacunar stroke may also occur in a patient with AF. Additional biomarkers are needed to identify high-risk individuals more accurately [27]. Second, systemic embolism related to AF occurs subsequent to thrombus generation in the cardiac chamber. To measure the individual risk of ischemic events more precisely, it would be necessary to consider the function of the cardiac chamber, atrial myopathy, duration and type of AF, serum and imaging biomarkers, genetic

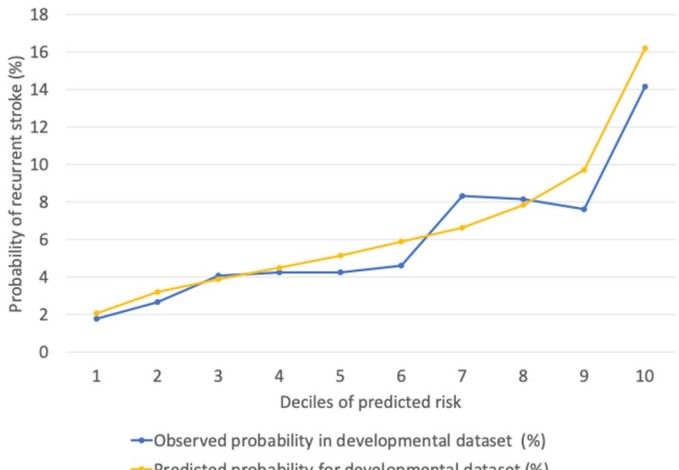

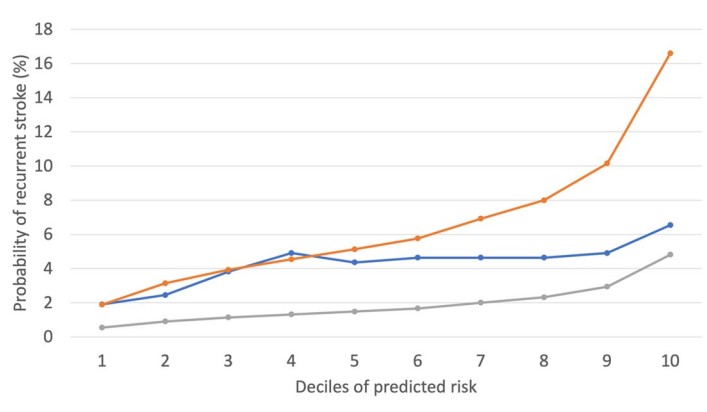

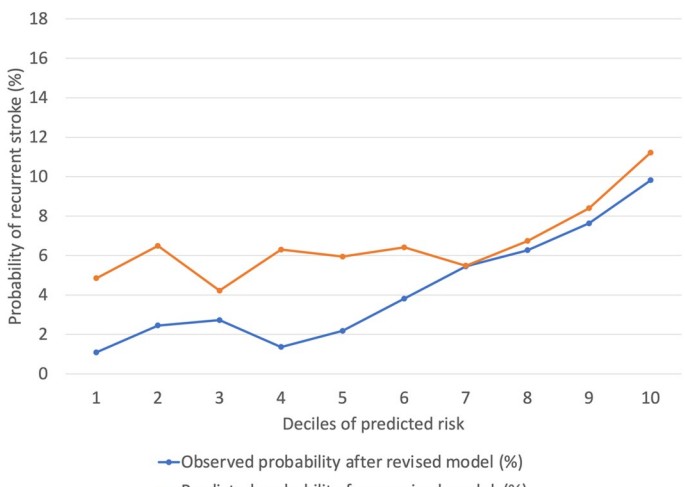

**Fig 2. Development, calibration, and revision processes of the prediction model.** The model's predictive ability was compared to the 1-year cumulative incidence of recurrent stroke as stratified by the deciles of predicted risks. A. Observed and predicted probability of recurrent stroke in the developmental dataset (N = 5648). B. Observed and predicted probability in the external validation dataset (N = 3668) before and after calibrating the model's overall slope. C. Observed and predicted probability in the external validation dataset (N = 3668) after revising the regression coefficients of the calibrated model.

predisposition, and so forth [28–31]. Third, improved medication adherence by introducing DOACs may mitigate the differential risk of recurrent stroke over the whole range of the risk scores [32]. Lastly, the number of recurrent stroke events in the developmental and validation datasets were relatively small, so that statistical power was not optimal.

A few points need further clarification. Our study was based on Korean and Japanese stroke populations; therefore, the generalizability of the study results to other races is uncertain. Japanese stroke patients have been reported to have a lower long-term mortality than that reported elsewhere in previous studies [33]. Because the final model incorporated non-stroke mortality as a competing risk for recurrent stroke, checking the reclassification performance of the conventional scores according to mortality was not feasible. Applying conventional risk scores to

**Table 2. Clinical prediction model for recurrent stroke among ischemic stroke patients with documented atrial fibrillation, incorporating the competing risk from all-cause mortality.**

| Variable | Coefficient | Standard error |
|---|---|---|
| Ischemic stroke versus TIA | 0.82 | 0.29 |
| Premorbid mRS ≥1 | 0.16 | 0.18 |
| Hypertension | 0.17 | 0.20 |
| Diabetes | 0.12 | 0.19 |
| History of ischemic heart disease | 0.40 | 0.22 |
| Smoking | -0.98 | 1.01 |
| Prestroke antiplatelet | 0.16 | 0.18 |
| Prestroke lipid-lowering medication | 0.31 | 0.19 |
| MCA or ICA occlusion | 0.80 | 0.35 |
| Basilar or vertebral artery occlusion | 1.04 | 0.41 |
| Number of arterial occlusion sites | -0.87 | 0.31 |
| Intravenous thrombolysis | 0.21 | 0.24 |
| Endovascular recanalization treatment | -0.23 | 0.24 |
| Discharge medication (use of DOAC) | 0.29 | 0.14 |
| Age | 0.01 | 0.01 |
| Square root of 1/BMI (per 0.01) | 0.13 | 0.06 |
| Square root of 1/Initial glucose (per 0.01) | -0.07 | 0.06 |
| Logarithm of Initial systolic blood pressure | 1.45 | 0.40 |
| Square root of 1/Platelet counts (per 0.01) | 0.01 | 0.02 |
| Logarithm of Prothrombin time | 1.20 | 0.72 |
| 1/(Initial NHISS score) | 0.48 | 0.47 |
| (Male sex) * (Smoking) | 0.93 | 1.03 |
| (Stroke subtype) * (Newly diagnosed atrial fibrillation) | -0.32 | 0.15 |
| (History of stroke or TIA) * (Intravenous thrombolysis) | 0.37 | 0.39 |
| (Logarithm of prothrombin time) * (Discharge medication) | -0.66 | 0.35 |
| (White blood cell count) * 1/(Initial NIHSS score) | -0.87 | 0.59 |

TIA, transient ischemic attack; mRS, modified Rankin Scale; MCA, middle cerebral artery; ICA, internal carotid artery; DOAC, direct oral anticoagulants; BMI, body mass index; NIHSS, National Institute of Health Stroke Scale.

**Table 3. Incidence rate stratified by the various risk scores.**

| ATRIA score | | | | | CHADS$_2$ score | | | | | CHA$_2$DS$_2$-VASc score | | | | | Newly developed model | | | | |
|---|---|---|---|---|---|---|---|---|---|---|---|---|---|---|---|---|---|---|---|
| Point | Event | 100-PY | Incidence Rate per 100-PY | OR (95% CI) | Point | Event | 100-PY | Incidence Rate per 100-PY | OR (95% CI) | Point | Event | 100-PY | Incidence Rate per 100-PY | OR (95% CI) | Decile* | Event | 100-PY | Incidence Rate per 100-PY | OR (95% CI) |
| 7 | 14 | 1.77 | 7.93 (4.70, 13.39) | Reference | 2 | 179 | 30.15 | 5.94 (5.13, 6.87) | Reference | 2 | 62 | 12.97 | 4.78 (3.73, 6.13) | Reference | 1 | 10 | 4.69 | 2.13 (1.15, 3.96) | Reference |
| 8 | 38 | 7.93 | 4.79 (3.48, 6.58) | 1.40 (0.70, 2.82) | 3 | 72 | 7.52 | 9.57 (7.60, 12.05) | 0.82 (0.60, 1.11) | 3 | 60 | 9.39 | 6.39 (4.96, 8.23) | 0.98 (0.66, 1.46) | 2 | 15 | 4.63 | 3.24 (1.95, 5.37) | 0.73 (0.32, 1.69) |
| 9 | 84 | 13.73 | 6.12 (4.94, 7.58) | 1.29 (0.67, 2.46) | 4 | 66 | 5.30 | 12.45 (9.78, 15.85) | 0.61 (0.44, 0.84) | 4 | 88 | 10.72 | 8.21 (6.66, 10.11) | 0.88 (0.61, 1.26) | 3 | 23 | 4.57 | 5.03 (3.34, 7.57) | 0.46 (0.21, 1.00) |
| 10 | 99 | 12.08 | 8.19 (6.73, 9.98) | 1.02 (0.53, 1.93) | 5 | 17 | 1.56 | 10.89 (6.77, 17.52) | 1.04 (0.60, 1.80) | 5 | 58 | 6.69 | 8.67 (6.70, 11.21) | 1.02 (0.68, 1.51) | 4 | 24 | 4.69 | 5.12 (3.43, 7.63) | 0.39 (0.18, 0.84) |
| 11 | 67 | 5.66 | 11.85 (9.32, 15.05) | 1.01 (0.52, 1.95) | 6 | 4 | 0.14 | 27.76 (10.42, 73.95) | 0.35 (0.10, 1.26) | 6 | 54 | 3.44 | 15.71 (12.04, 20.52) | 0.53 (0.35, 0.81) | 5 | 24 | 4.63 | 5.18 (3.47, 7.73) | 0.40 (0.18, 0.87) |
| 12 | 28 | 2.54 | 11.00 (7.60, 15.94) | 1.28 (0.62, 2.67) | | | | | | 7 | 10 | 1.26 | 7.92 (4.26, 14.72) | 1.46 (0.71, 3.01) | 6 | 26 | 4.54 | 5.72 (3.90, 8.41) | 0.39 (0.18, 0.85) |
| 13 | 7 | 0.76 | 9.24 (4.40, 19.38) | 1.95 (0.71, 5.32) | | | | | | 8 | 6 | 0.19 | 31.67 (14.23, 70.51) | 0.28 (0.09, 0.83) | 7 | 47 | 4.42 | 10.63 (7.99, 14.15) | 0.21 (0.10, 0.43) |
| 14 | 1 | 0.17 | 6.05 (0.85, 42.99) | 4.14 (0.50, 34.61) | | | | | | 9 | 0 | - | - | - | 8 | 46 | 4.28 | 10.74 (8.04, 14.33) | 0.24 (0.11, 0.49) |
| 15 | 0 | - | - | - | | | | | | | | | | | 9 | 43 | 4.41 | 9.76 (7.24, 13.16) | 0.24 (0.11, 0.49) |
| | | | | | | | | | | | | | | | 10 | 80 | 3.80 | 21.03 (16.89, 26.19) | 0.16 (0.08, 0.33) |

* The estimated individual probability of recurrent stroke was categorized into deciles for the purpose of comparison.

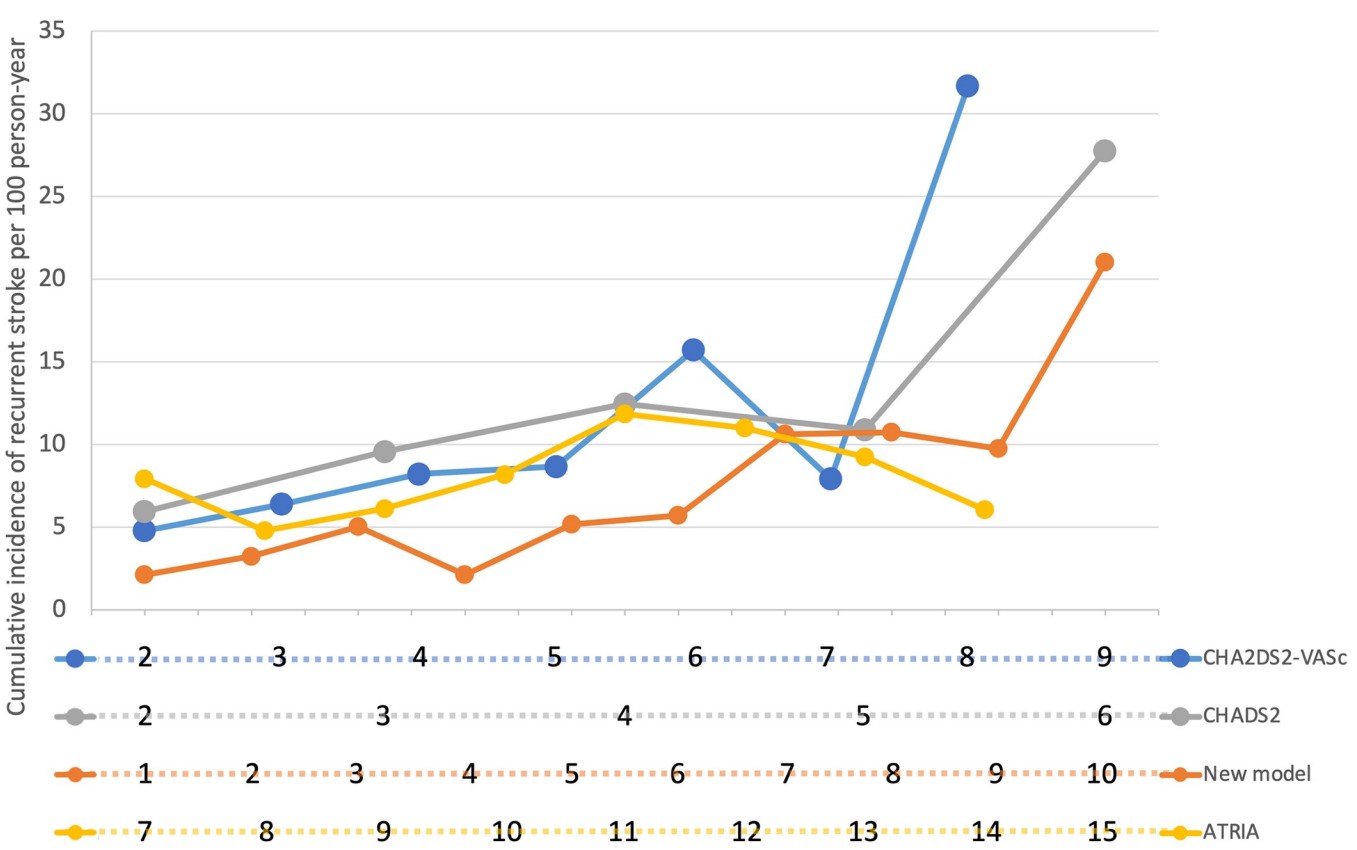

**Fig 3. Cumulative incidence of recurrent stroke by the various risk scores.** *A CHA$_2$DS$_2$-VASc score of 9 points and an ATRIA score of 15 points were not observed in the dataset. The estimated individual probability of recurrent stroke was categorized into deciles for the purpose of comparison.

the AIS population was beyond the intention of developing these scores. The number of DOAC prescriptions rapidly increased in Korea after 2015, when it was approved for the reimbursement list. Thus, the proportion of DOAC usage increased to 49% in the external validation set from 10% in the development set.

## Conclusion

We developed and validated a comprehensive risk prediction model for recurrent stroke in East Asian patients with ischemic stroke and non-valvular AF. The newly developed model showed only modest utility in discriminating the risk of recurrence, similar to the conventional risk scores (ATRIA, CHADS$_2$, and CHA$_2$DS$_2$-VASc scores). Detailed individual information, including brain imaging, serum biomarkers, and cardiac function, may be needed to build a more robust and precise risk prediction model.

## Supporting information

**S1 Fig. Distribution of CHADS2, CHA2DS2-VASc, and ATRIA scores in the developmental dataset.**
(PDF)

**S2 Fig. Distribution of CHADS2, CHA2DS2-VASc, and ATRIA scores in the developmental dataset by countries.**
(PDF)

**S1 Table. Clinical profile of the external validation dataset.**
(PDF)

**S1 File. Development, recalibration, and revision processes of the prediction model.** The model's predictability was compared to the observed probability of recurrent stroke stratified by the deciles of predicted risks.
(PDF)

## Acknowledgments

### List of participating researchers in the CRCS-K

Hee-Joon Bae, Beom Joon Kim, Moon-Ku Han, Jihoon Kang, Jun Yup Kim, Keon-Joo Lee (Seoul National University Bundang Hospital, Seongnam); Jun Lee, Doo Hyuk Kwon (Yeungnam University Medical Center, Daegu); Jee-Hyun Kwon, Wook-Joo Kim (Ulsan University Hospital, Ulsan); Jae-Kwan Cha, Dae-Hyun Kim, Jin-Heon Jeong (Dong-A University Hospital, Busan); Jay Chol Choi, Joong-Goo Kim, Chul Hoo Kang (Jeju National University Hospital, Jeju); Joon-Tae Kim, Ki-Hyun Cho, Man-Seok Park, Kang-Ho Choi (Chonnam National University Hospital, Gwangju); Sung-Il Sohn, Jeong-Ho Hong, Hyungjong Park (Keimyung University Dongsan Medical Center, Daegu); Soo Joo Lee, Jae Guk Kim (Eulji University Hospital, Daejeon); Dong-Ick Shin, Kyu Sun Yum: Baik-Kyun Kim (Chungbuk National University Hospital, Cheongju); Tai Hwan Park, Sang-Soon Park (Seoul Medical Center, Seoul); Byung-Chul Lee, Mi-Sun Oh, Kyung-Ho Yu, Minwoo Lee (Hallym University Sacred Heart Hospital, Anyang); Kyung Bok Lee (Soon Chun Hyang University Hospital Seoul, Seoul); Keun-Sik Hong, Yong-Jin Cho, Hong-Kyun Park (Inje University Ilsan Paik Hospital, Goyang); Dong-Eog Kim, Wi-Sun Ryu (Dongguk University Ilsan Hospital, Goyang); Jong-Moo Park, Kyusik Kang, Inyoung Chung (Eulji General Hospital, Seoul); Chulho Kim, Sang-Hwa Lee (Hallym University Chuncheon Sacred Heart Hospital, Chuncheon); Kwang Yeol Park, Hae-Bong Jeong (Chung-Ang University Hospital, Seoul);

Juneyoung Lee, PhD (Department of Biostatistics, Korea University, Seoul)

### List of participating researchers in the SAMURAI-NVAF study

Kenichi Todo (Kobe City Medical Center General Hospital, Kobe, Japan); Yoshiki Yagita, Kazumi Kimura, Kensaku Shibasaki (Kawasaki Medical School, Kawasaki); Ryo Itabashi, Eisuke Furui (Kohnan Hospital, Sendai); Tadashi Terasaki (Japanese Red Cross Kumamoto Hospital, Kumamoto); Yoshiaki Shiokawa, Teruyuki Hirano, Rieko Suzuki (Kyorin University School of Medicine, Mitaka); Kenji Kamiyama, Jyoji Nakagawara (Nakamura Memorial Hospital, Sapporo); Shunya Takizawa, Kazunari Homma (Tokai University School of Medicine, Kanagawa); Satoshi Okuda (NHO Nagoya Medical Center, Nagoya); Yasushi Okada, Koichiro Maeda (NHO Kyushu Medical Center, Fukuoka); Tomoaki Kameda, Kazuomi Kario (Jichi Medical University School of Medicine, Tochiki); Yoshinari Nagakane (Kyoto Second Red Cross Hospital, Kyoto); Yasuhiro Hasegawa, Hisanao Akiyama (St. Marianna University School of Medicine, Kawasaki); Satoshi Shibuya, Hiroshi Mochizuki (South Miyagi Medical Center, Miyagi); Yasuhiro Ito (TOYOTA Memorial Hospital, Toyota); Hideki Matsuoka, Takahiro Nakashima (NHO Kagoshima Medical Center, Kagoshima); Kazuhiro Takamatsu (Brain Attack Center Ota Memorial Hospital, Hiroshima); Kazutoshi Nishiyama (Kitasato University School of Medicine, Kanagawa); Kazunori Toyoda, Masatoshi Koga, Sohei Yoshimura, Kanta Tanaka, Shoji Arihiro, Masayuki Shiozawa (National Cerebral and Cardiovascular Center, Suita)

## Author Contributions

**Conceptualization:** Hee-Joon Bae.

**Data curation:** Beom Joon Kim, Keon-Joo Lee, Kanta Tanaka, Masatoshi Koga, Sohei Yoshimura, Ryo Itabashi, Jae-Kwan Cha, Byung-Chul Lee, Hisanao Akiyama, Yoshinari Nagakane, Kazunori Toyoda, Hee-Joon Bae.

**Formal analysis:** Beom Joon Kim, Eun Lyeong Park, Juneyoung Lee.

**Funding acquisition:** Kazunori Toyoda, Hee-Joon Bae.

**Methodology:** Beom Joon Kim, Eun Lyeong Park, Juneyoung Lee.

**Supervision:** Hee-Joon Bae.

**Validation:** Juneyoung Lee.

**Visualization:** Beom Joon Kim.

**Writing – original draft:** Beom Joon Kim.

**Writing – review & editing:** Keon-Joo Lee, Kanta Tanaka, Masatoshi Koga, Sohei Yoshimura, Ryo Itabashi, Jae-Kwan Cha, Byung-Chul Lee, Hisanao Akiyama, Yoshinari Nagakane, Juneyoung Lee, Kazunori Toyoda, Hee-Joon Bae.

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
