## [Decision Letter · Decision Letter 0]

29 Jun 2021

PONE-D-21-16999

Prediction of recurrent stroke among ischemic stroke patients with atrial fibrillation: Development and validation of a risk score model

PLOS ONE

Dear Dr. Kim,

Thank you for submitting your manuscript to PLOS ONE. After careful consideration, we feel that it has merit but does not fully meet PLOS ONE’s publication criteria as it currently stands. Therefore, we invite you to submit a revised version of the manuscript that addresses the points raised during the review process.

We look forward to receiving your revised manuscript.

Kind regards,

Adam Wiśniewski

Academic Editor

PLOS ONE

Journal Requirements:

2. Thank you for including your ethics statement: "The local institutional review boards (IRBs) of all participating centers approved the study with a waiver of consent. Secondary use of the registry data and additional review of medical records for this study were approved by IRBs [B-1705/396-306]."

4. Thank you for stating the following in the Financial Disclosure section:

"This study was supported by Bristol‒Myers Squibb Korea and the Korea Centers for

Disease Control and Prevention (no. 2020ER620200#) granted to HJB and a Grant-in-

Aid (H23-Junkanki-Ippan-010) from the Ministry of Health, Labour and Welfare, Japan

granted to KT. The funding sources did not participate in any part of the study, from

conception to article preparation."

We note that you received funding from a commercial source: "Bristol‒Myers Squibb"

7. One of the noted authors is a group or consortium "SAMURAI Study Investigators and CRCS-K Investigators". In addition to naming the author group, please list the individual authors and affiliations within this group in the acknowledgments section of your manuscript. Please also indicate clearly a lead author for this group along with a contact email address.

Additional Editor Comments (if provided):

I am sorry but we are not able to process the paper in the present form.

If You feel You can extensively modify and significantly improve Your paper according to the Reviewer's criticism - we give You a chance for a major revision.

Reviewers' comments:

Reviewer's Responses to Questions

**Comments to the Author**

1. Is the manuscript technically sound, and do the data support the conclusions?

Reviewer #1: Yes

Reviewer #2: Partly

2. Has the statistical analysis been performed appropriately and rigorously? 

Reviewer #1: Yes

Reviewer #2: Yes

3. Have the authors made all data underlying the findings in their manuscript fully available?

Reviewer #1: Yes

Reviewer #2: Yes

4. Is the manuscript presented in an intelligible fashion and written in standard English?

Reviewer #1: Yes

Reviewer #2: Yes

5. Review Comments to the Author

Reviewer #1: This study aimed to develop and test a risk prediction model for recurrent stroke in patients with AF who had experienced an initial stroke. The sample size was sufficient and the methods were sound for answering this question. The authors found that the risk prediction model was not superior to current models of CHADS2 and CHA2DS2-VASC and ATRIA. The authors make reasonable conclusions regarding the need for additional measures/imaging to improve recurrent stroke prediction in these patients.

The paper validates the tools in an East Asian Population (Korea and Japan) and therefore this does add to the body of evidence on this topic. The paper is presented in a logical manner and is well written. I have no major comments to make on the paper.

Reviewer #2: The Authors tried to develop a comprehensive risk prediction model for recurrent stroke from non-valvular AF stroke patients in Korea and Japan registry database. The model was subsequently externally-validated in another dataset, also from Korea/Japan registry. After carefully validated, both internally and externally, the final model included 21 variables as well as several interaction terms. However, the performance of this model only reached modest performance with unsatisfactory discriminative ability among patients with middle score range, which could not surpass the currently-used popular risk score systems. Since current prediction model contains numerous detailed variables and its application would depend on solid electric medical records and programmed automated calculation, the clinical feasibility could be low under not-outstanding prediction performance.

Furthermore, I have some other concerns:

1. There is a crucial diversity between the developmental and validated cohort: the percentage of DOAC use. In the model (Table 2), DOAC use at discharge only revealed 0.29 coefficient, maybe partly related to the low use rate of the developmental cohort. However, years after, in the validated cohort, the percentage of DOAC use abruptly increased. The Authors also addressed this issue and presumed the stroke risk modification in the DOAC era. Hence, the developmental dataset may need to be modified, eg.: using 2015-2016 stroke cohort for further model development.

2. The model was built under "Fine-Gray sub-distribution hazard model". Though reasonable in current risk competing model, the final event estimation rate would be lower, especially the high 1-year mortality rate (16%) was noted in the cohort. This might also affect the model performance.

3. The rating score 1-10 was not appropriately described. Readers could only tell the numerous comprehensive predictors listed in Table 2 and their coefficients. The Authors did not explain how these weighted parameters transformed into 1-10 scores.

6. PLOS authors have the option to publish the peer review history of their article (what does this mean?). If published, this will include your full peer review and any attached files.

Reviewer #1: No

Reviewer #2: No

---

## [Author Response · Author response to Decision Letter 0]

20 Aug 2021

Rebuttal to the reviewers' comment

PLoS One

Manuscript ID, PONE-D-21-16999R1

Manuscript title: Prediction of recurrent stroke among ischemic stroke patients with atrial fibrillation: Development and validation of a risk score model

Aug 21, 2021

Dear reviewers, Dr. Adam Wiśniewski, editorial staff, and Dr. Emily Chenette

 We thank you and the referees of PLoS One for taking the time to review our manuscript and giving us a chance to revise the paper. Our responses follow the itemized list of reviewers' and editor's original comments. We are thankful to reviewers and editors for their constructive and helpful suggestions, and we tried our best to meet the scientific and literary level of PLoS One. We also followed all the requests from the editorial office on August 20, 2021. Additionally, we modified several sentences to clarify our point, some of which were not requested by the reviewers and not specified in the rebuttal. The marked version contains all the changes from the previous version of our manuscript.

EDITORIAL COMMENTS: 

Reply to 1)

We read the journal style requirements and followed them as requested.

2. Thank you for including your ethics statement: "The local institutional review boards (IRBs) of all participating centers approved the study with a waiver of consent. Secondary use of the registry data and additional review of medical records for this study were approved by IRBs [B-1705/396-306]."

Once you have amended this/these statement(s) in the Methods section of the manuscript, please add the same text to the "Ethics Statement" field of the submission form (via "Edit Submission").

Reply to 2)

We specified the name of the institution that has given the final approval of this study. The list of all participating centers is provided in the Acknowledgement.

Secondary use of the registry data and additional review of medical records for this study were approved by the IRB of Seoul National University Bundang Hospital [B-1705/396-306].

3. We note that you have indicated that data from this study are available upon request. PLOS only allows data to be available upon request if there are legal or ethical restrictions on sharing data publicly.

Reply to 3)

We concur with the principle of data sharing led by the PLoS group. Unfortunately, however, we have to write that the source data could not be made publicly available due to legal constraints, specifically the Personal Information Protection Act (2014). No explicit informed consent for public archiving of the pseudonymized source data has been obtained, in which case local regulations preclude public archiving of the data. Pseudonymized data that support the findings of this study are available from the corresponding author, Dr. Hee-Joon Bae, upon request, subsequent approval from the Institutional Review Board of the Seoul National University Bundang Hospital, and completion of a legal data sharing agreement.

Link to the full text of the Personal Information Protection Act: https://www.law.go.kr/LSW/lsInfoP.do?chrClsCd=010203&lsiSeq=142563&viewCls=engLsInfoR&urlMode=engLsInfoR#0000

4. Thank you for stating the following in the Financial Disclosure section:

"This study was supported by Bristol‒Myers Squibb Korea and the Korea Centers for

Disease Control and Prevention (no. 2020ER620200#) granted to HJB and a Grant-in-

Aid (H23-Junkanki-Ippan-010) from the Ministry of Health, Labour and Welfare, Japan granted to KT. The funding sources did not participate in any part of the study, from

conception to article preparation."

We note that you received funding from a commercial source: "Bristol‒Myers Squibb"

Within this Competing Interests Statement, please confirm that this does not alter your adherence to all PLOS ONE policies on sharing data and materials by including the following statement: "This does not alter our adherence to PLOS ONE policies on sharing data and materials." (as detailed online in our guide for authors http://journals.plos.org/plosone/s/competing-interests). If there are restrictions on sharing of data and/or materials, please state these. Please note that we cannot proceed with consideration of your article until this information has been declared.

Reply to 4)

Thank you for pointing out this. We amended the Competing Interests Statement as indicated. 

Reply to 5)

This point is duly noted. The competing interests statements were changed along with point #4. 

Reply to 6)

We read the journal style requirements and followed as indicated.

7. One of the noted authors is a group or consortium "SAMURAI Study Investigators and CRCS-K Investigators". In addition to naming the author group, please list the individual authors and affiliations within this group in the acknowledgments section of your manuscript. Please also indicate clearly a lead author for this group along with a contact email address.

Reply to 7)

We explicitly stated lead authors for the two study groups and indicated the complete list of participating researchers in the acknowledgments section.

Reviewer #1:

Reviewer #1: This study aimed to develop and test a risk prediction model for recurrent stroke in patients with AF who had experienced an initial stroke. The sample size was sufficient and the methods were sound for answering this question. The authors found that the risk prediction model was not superior to current models of CHADS2 and CHA2DS2-VASC and ATRIA. The authors make reasonable conclusions regarding the need for additional measures/imaging to improve recurrent stroke prediction in these patients.

The paper validates the tools in an East Asian Population (Korea and Japan) and therefore this does add to the body of evidence on this topic. The paper is presented in a logical manner and is well written. I have no major comments to make on the paper.

Reply to Reviewer #1)

We appreciate Reviewer #1's positive evaluation of our study.

Reviewer #2:

Reviewer #2: The Authors tried to develop a comprehensive risk prediction model for recurrent stroke from non-valvular AF stroke patients in Korea and Japan registry database. The model was subsequently externally-validated in another dataset, also from Korea/Japan registry. After carefully validated, both internally and externally, the final model included 21 variables as well as several interaction terms. However, the performance of this model only reached modest performance with unsatisfactory discriminative ability among patients with middle score range, which could not surpass the currently-used popular risk score systems. Since current prediction model contains numerous detailed variables and its application would depend on solid electric medical records and programmed automated calculation, the clinical feasibility could be low under not-outstanding prediction performance.

Reply to the critique.

We appreciate reviewer #2's thoughtful and constructive points. We concur with the critic that our model was not superior to the conventional scores and, in short of its clinical feasibility due to the use of extensive clinical information. 

Based on the newly developed model's performance, we could infer the following points;

1) However extensive stroke physicians have gathered clinical variables, it is still incapable of accurately predicting stroke survivors' prognosis. 

2) The performance of conventional models, including CHADS2, CHA2DS2-VASc, and ATRIA scores, was only par with our model, showing unsatisfactory prediction of recurrent ischemic stroke. 

In short, by developing an extensive risk prediction model and comparing it to conventional scores, we concluded that further mechanistic information is required to build a more robust and precise risk prediction model, such as brain imaging, serum biomarkers, and cardiac function.

Furthermore, I have some other concerns:

1. There is a crucial diversity between the developmental and validated cohort: the percentage of DOAC use. In the model (Table 2), DOAC use at discharge only revealed 0.29 coefficient, maybe partly related to the low use rate of the developmental cohort. However, years after, in the validated cohort, the percentage of DOAC use abruptly increased. The Authors also addressed this issue and presumed the stroke risk modification in the DOAC era. Hence, the developmental dataset may need to be modified, eg.: using 2015-2016 stroke cohort for further model development.

Reply to 1)

We agree with the reviewer's point. After the efficacy of DOACs was proven through randomized clinical trials around 2011, the widespread use of DOACs began after 2015, when the reimbursement of DOACs by the Korean government was started. Thus, it was impossible to gather patients with atrial fibrillation taking vitamin K antagonists instead of DOACs in Korea and Japan. Additionally, DOAC became the treatment of choice for the prevention of recurrent ischemic events after stroke. Therefore, it is more appropriate to validate the estimated recurrent stroke risk in patients with DOACs. Lastly, most clinical trials reported the comparable efficacy of VKA and DOAC, which also made it reasonable to validate the developed model in a dataset of higher DOAC usage. 

Regarding the above discussion, we added the following sentence to the discussion section; Thus, the proportion of DOAC usage increased to 49% in the external validation set from 10% in the development set.

2. The model was built under "Fine-Gray sub-distribution hazard model". Though reasonable in current risk competing model, the final event estimation rate would be lower, especially the high 1-year mortality rate (16%) was noted in the cohort. This might also affect the model performance.

Reply to 2)

We thank the reviewer for pointing this out.

The mortality rate of ischemic stroke patients with atrial fibrillation varies by geographic and demographic characteristics. A pooled analysis of seven prospective cohort studies reported a mortality rate of around 10% per year with oral anticoagulation. [Seiffge. Ann Neurol. 2020] From a health maintenance organization database in the US, the 1-year mortality after ischemic stroke with atrial fibrillation was reported to be 40%; however, the average age of included patients was higher than that of our cohort, the US database included a higher prevalence of cancer history, and the treatment information was missing in the study. [Fang. Neurology. 2014] 

In a cohort with a relatively high mortality rate, it is required to consider the effect of competing risk. Estimation of incidence in the presence of competing risks is known to be biased upwards if the competing event, i.e., mortality for our study, is ignored. [Austin P. Stat Med. 2016]

Abdel-Qadir et al. reported that the stroke incidence was overestimated by a relative factor of 39% when the Fine-Gray model was not applied to an atrial fibrillation cohort from the Ontario provincial health care database. [Abdel-Qadir. Circ Cardiovasc Qual Outcome. 2018]

Therefore, we consider the Fine-Gray model is appropriate for estimating recurrent stroke risk from ischemic stroke patients with atrial fibrillation. 

3. The rating score 1-10 was not appropriately described. Readers could only tell the numerous comprehensive predictors listed in Table 2 and their coefficients. The Authors did not explain how these weighted parameters transformed into 1-10 scores.

Reply to 3)

We appreciate the reviewer for letting us elaborate on this. 

The output from our model is not a score ranging from 1 to 10, but an individual probability of having recurrent stroke ranging from 0 to 1. We categorized the individual probability into deciles to compare it with traditional scoring systems. 

We added the following sentence as a caption after Table 3 and Figure 2. 

The estimated individual probability of recurrent stroke was categorized into deciles for the purpose of comparison.

Additional edits requested from the editorial office on August 20, 2021

1. Please upload a Response to Reviewers letter which should include a point by point response to each of the points made by the Editor and / or Reviewers. (This should be uploaded as a 'Response to Reviewers' file type.) Please follow this link for more information: http://blogs.PLOS.org/everyone/2011/05/10/how-to-submit-your-revised-manuscript/

Reply to 1)

We uploaded the document file. 

"This study was supported by Bristol‒Myers Squibb Korea and the Korea Centers for Disease Control and Prevention (no. 2020ER620200#) and a Grant-in-Aid (H23-Junkanki-Ippan-010) from the Ministry of Health, Labour and Welfare, Japan. This does not alter our adherence to PLOS ONE policies on sharing data and materials. The funding sources did not participate in any part of the study, from conception to article preparation. "

Please include your amended statements within your cover letter; we will change the online submission form on your behalf."

Reply to 2)

We deleted the funding information from the acknowledgement section. The statement is added to the cover letter.

3. Please upload a new copy of Figure 2 as the detail is not clear. Please follow the link for more information: https://blogs.plos.org/plos/2019/06/looking-good-tips-for-creating-your-plos-figures-graphics/

Reply to 3)

We increased the font size of Figure 2 and uploaded it. 

4. Please provide additional details regarding participant consent. In the Methods section, please ensure that you have specified (1) whether consent was informed and (2) what type you obtained (for instance, written or verbal). If your study included minors, state whether you obtained consent from parents or guardians. If the need for consent was waived by the ethics committee, please include this information.

Reply to 4)

All the participants had given their written informed consents when they participated in the original CRCS-K and SAMURAI-NVAF studies. For the current post-hoc study, written consents were waived by the institutional review board of the Seoul National University Bundang Hospital. This information was cleared mentioned in the method section.

5. Thank you for updating your Data Availability statement which we note as the following: "The pseudonymized data that support the findings of this study are available from the corresponding author, Dr. Hee-Joon Bae, upon reasonable request, subsequent approval from the local IRB, and completion of a legal data sharing agreement.

We concur with the principle of data sharing led by the PLoS group. Unfortunately, however, we have to write that the source data could not be made publicly available due to legal constraints, specifically the Personal Information Protection Act (2014). No explicit informed consent for public archiving of the pseudonymized source data has been obtained, in which case local regulations preclude public archiving of the data. Pseudonymized data that support the findings of this study are available from the corresponding author, Dr. Hee-Joon Bae, upon request, subsequent approval from the Institutional Review Board of the Seoul National University Bundang Hospital, and completion of a legal data sharing agreement.

Link to the full text of the Personal Information Protection Act: https://www.law.go.kr/LSW/lsInfoP.do?chrClsCd=010203&lsiSeq=142563&viewCls=engLsInfoR&urlMode=engLsInfoR#0000"

PLOS journals require authors to make all data necessary to replicate their study’s findings publicly available without restriction at the time of publication. When specific legal or ethical restrictions prohibit public sharing of a data set, authors must indicate how others may obtain access to the data.

Thank you for providing information regarding the ethical restrictions for your data. Please provide contact information for the ethics committee or governing body mandating this restriction. Please note it is not acceptable for an author to be the sole named individual responsible for ensuring data access. We hope to hear from you soon.

Reply to 5) 

We amended the section as following, adding highlighted sentences; 

Secondary use of the registry data and additional review of medical records for the current study were approved by the IRB of Seoul National University Bundang Hospital [B-1705/396-306]. The source data could not be made publicly available due to legal constraints, specifically the Personal Information Protection Act (2014). No explicit informed consent for public archiving of the pseudonymized source data has been obtained, in which case local regulations preclude public archiving of the data. The pseudonymized data that support the findings of this study are available from the corresponding author, Dr. Hee-Joon Bae, or the IRB of Seoul National University Bundang Hospital (82, Gumi-ro 173 Beon-gil, Bundang-gu, Seongnam-si, Gyeonggi-do 13605, South Korea; https://msri.snubh.org) upon reasonable request, subsequent approval from the local IRB, and completion of a legal data sharing agreement.

---

## [Decision Letter · Decision Letter 1]

27 Sep 2021

Prediction of recurrent stroke among ischemic stroke patients with atrial fibrillation: 

Development and validation of a risk score model

PONE-D-21-16999R1

Dear Dr. Kim,

We’re pleased to inform you that your manuscript has been judged scientifically suitable for publication and will be formally accepted for publication once it meets all outstanding technical requirements.

Kind regards,

Adam Wiśniewski

Academic Editor

PLOS ONE

Additional Editor Comments (optional):

Reviewers' comments:

Reviewer's Responses to Questions

**Comments to the Author**

1. If the authors have adequately addressed your comments raised in a previous round of review and you feel that this manuscript is now acceptable for publication, you may indicate that here to bypass the “Comments to the Author” section, enter your conflict of interest statement in the “Confidential to Editor” section, and submit your "Accept" recommendation.

Reviewer #3: All comments have been addressed

2. Is the manuscript technically sound, and do the data support the conclusions?

Reviewer #3: Yes

3. Has the statistical analysis been performed appropriately and rigorously? 

Reviewer #3: Yes

4. Have the authors made all data underlying the findings in their manuscript fully available?

Reviewer #3: Yes

5. Is the manuscript presented in an intelligible fashion and written in standard English?

Reviewer #3: Yes

6. Review Comments to the Author

Reviewer #3: Comments to the Author

The authors present developmental series consisted of 4483 Korean and 1165 Japanese patients as to risk prediction model for recurrent events among patients with acute ischemic stroke (AIS) and atrial fibrillation (AF). The patient population in the present study is large. In this study, 338 patients (6%) had recurrent stroke. The authors found there were no statistically significant differences between the newly developed model and the conventional risk scores (ATRIA, CHADS2, and CHA2DS2-VASc scores) assessing recurrence risk. The data and methods may certainly be of use for neurologists and strokologists.

Overall, this manuscript is of a potential interest to the clinicians. I believe the paper will be of interest to the readership of PLOS ONE and would recommend it for acceptance

7. PLOS authors have the option to publish the peer review history of their article (what does this mean?). If published, this will include your full peer review and any attached files.

Reviewer #3: No

---

## [Editor Report · Acceptance letter]

30 Sep 2021

PONE-D-21-16999R1 

Prediction of recurrent stroke among ischemic stroke patients with atrial fibrillation: 
Development and validation of a risk score model 

Dear Dr. Kim:

I'm pleased to inform you that your manuscript has been deemed suitable for publication in PLOS ONE. Congratulations! Your manuscript is now with our production department. 

Kind regards, 

on behalf of

Dr. Adam Wiśniewski 

Academic Editor

PLOS ONE